# State Estimation in a Biodigester via Nonlinear Logistic Observer: Theoretical and Simulation Approach

Abraham Efraím Rodríguez-Mata [1], Emanuel Gómez-Vidal [2], Carlos Alexander Lucho-Constantino [3], Jesús A. Medrano-Hermosillo [1], Rogelio Baray-Arana [1] and Pablo A. López-Pérez [4,*]

1 División de Estudios de Posgrado e Investigación, Tecnologico Nacional de Mexico Campus Chihuahua, Chihuahua 31310, Mexico; abraham.rm@chihuahua.tecnm.mx (A.E.R.-M.)
2 CITIS-ICBI, Autonomous Hidalgo State University, Pachuca 42184, Mexico
3 Área Académica de Química, Universidad Autónoma del Estado de Hidalgo, Pachuca de Soto 42039, Mexico
4 Escuela Superior de Apan, Autonomous University of Hidalgo State, Apan 43900, Mexico
* Correspondence: save1991@yahoo.com.mx

**Abstract:** The state variables in a biodigester are predicted using an unstructured model, and this study offers an analytical design of a Non-Linear Logistic Observer (NLLO), subsequently comparing its performance to that of other prominent state estimators. Because of variables such as temperature, pH, high pressure, volumetric organic load (VOC), and hydraulic retention time (HRT), among others, biodigester samples can be affected by the use of physical sensors, which are not always practical owing to their sensitivity to the type of sampling and external disturbances. The use of virtual sensors represents one approach to solving this issue. In this work, we suggest experimentally validating a mathematical model, then analytically designing a novel NLLO observer, and finally comparing the results to those obtained using a sliding-mode estimator and a Luenberger observer. By including online $CH_4$ and $CO_2$ measurements as inputs to the proposed observer, the local observability analysis demonstrated that all state variables were recoverable. After showing how well the suggested observer performs in numerical experiments, a proof based on the Lyapunov theory is offered. The primary innovation of this study is the incorporation of a novel algorithm that has been empirically validated and has output resilience to input parametric perturbations.

**Keywords:** observers; bioreactor; virtual sensors

## 1. Introduction

Anaerobic digestion (AD) is a natural fermentation process that takes place in the absence of oxygen. Microorganisms (biomass) turn biodegradable material (substrate) into a mixture of gases (usually methane and carbon dioxide) called biogas [1]. In addition, the degradation of organic matter offers a nutrient-rich energy source for microorganisms, generating compounds, such as biogas and biofertilizer, that are suitable for use in a biorefinery system. The rising need for fossil fuels has prompted the exploration of renewable energy sources, with the AD process emerging as a frontrunner due to the lower initial investment required to put increasingly into practice compared to other renewable energy sources such as wind, solar, and hydro [2]. Wastewater, sewage sludge, organic waste from the agro-food sector, and concrete urban debris can all be treated using this method [3]. Biogas is a flexible renewable energy source since methane and hydrogen may be utilized as a raw material in energy production. This means it can be used to generate electricity, heat buildings, and power vehicles [4].

Multiple methods, such as state estimators, are utilized to monitor and maintain the smooth operation of AD processes. Refs. [5,6], combined state estimator and physical sensors to monitor vital process variables constituting what are referred to as "virtual sensors" [5]. Superior automation and control systems in tandem with cutting-edge measuring equipment are necessary for implementing novel control schemes that will ensure the optimal and stable stabilization of AD operations [7–9]. Existing monitoring equipment for AD process essential

factors is costly and requires frequent maintenance. These variables include, but are not limited to, fatty acid concentrations, the central bacterial population, and the measurement of gases generated [10,11]. Few variables, including pH, temperature, biomass, and the ratio of select gases, are often available for online monitoring at a reasonable cost in an AD process. The use of so-called state observers (software sensors) that have demonstrated the ability to reconstruct state variables represents an exciting alternative that takes advantage of the mathematical model in conjunction with a limited number of measures in line, that can provide an estimate of the evolution over time of the state of the bioprocess [12]. Over the last few decades, the design and application of state observers in bioprocesses has been an area of interest [13]. Specifically for AD processes, in the literature, we can find different state estimation schemes from classic Kalman filters and adaptive to observer schemes asymptotic and interval [14]. There is currently no solution in the literature for the challenging issue regarding the online estimatation of critical variables in processes AD when only the biogas output flow rate is available for mediation. There are limited examples of AD estimating systems for tracking biogas production in the published literature; to estimate these processes, we employ observers of state with the following features:

1.  Exponential observers: These are based on the process model and use some measurements from a hardware sensor; the main disadvantage of these observers is that they depend heavily on the process model quality [15]. The extended Luenberger observer [16], the receding horizon observer, and the high-gain observers [17] are exponential observers.
2.  Asymptotic observers: Unlike exponential observers, the Asymptotic observers are open-loop state estimators, which use only a part of the model, replacing the missing piece of a model by a variable measure [18,19].
3.  Hybrid observers: The basic idea behind hybrid observers is to combine the advantages of exponential and asymptotic observers [20].

Other important observers include the so-called continuous discrete observers, which have shown to be extremely adaptable and resilient when the output signals are in discrete mode as robots or quadrotors [21]. Our main real contribution in the observer is the ability to estimate state variables that are usually not measurable, estimated via the signals that are on line; this is an excellent contribution for colleagues who are committed to the control and study of fermentation processes in bioreactors and biodigesters; by programming this observer in real-time or off-line tasks, they will be able to have a better follow-up of the dynamics of the various states of interest. A continuous NLLO observer is presented in this paper for use with an analog output sensor; however, its discrete extension for discrete sensors may be explored in future work. It can be difficult and costly to equip an organization for AD procedures with the appropriate instrumentation [7,22].

Research on the design of estimators across their many classes is highly relevant to the estimate of immeasurable variables in fermenters and biodigesters, and hence informs this study [23–27]. As a result, it strives to do away with the prohibitive monetary expense of installing such machinery in AD facilities. Some system state variables can be reconstructed by means of a virtual sensor (Soft-sensing) utilizing data from a real sensor. Because of this, it can be studied and applied to lessen the financial burden of biodigester substrate, biogas, and biomass sampling. Industrial plants need for accurate readings in a biodigester makes it crucial to choose which estimator has a sufficient performance. In this research, we offer a new Non-Linear Logistic Observer based nonlinear of type functions; this observer provides feedback on the nonlinear error. The Sliding Mode Observer and Luenberger Observer, two linear observers, are also contrasted. The proposed observer is further supported by a proof based on Lyapunov Stability. At last, a simulated input disturbance and quantifiable signal noise are used to prove the robustness of the observer.

## 2. Mathematical Model an Statement Problem

Anaerobic digesters are capable of treating insoluble waste and wastewater soluble. HRT of at least 10–20 days are typical for high-strength residues. High-speed anaerobic digesters are used for the treatment of soluble wastewater. A biodigester is hermetically sealed and waterproof, inside which organic matter such as manure and vegetable waste is deposited (being careful with the mixing of the substrates due to the substrate acidification). In a biodigester, the organic matter is fermented with a certain amount of water, producing methane gas and organic fertilizers rich in phosphorus, potassium, and nitrogen. This system can also include a loading chamber, a device to capture and store biogas, and hydraulic pumping, and posttreatment chambers (decanters, filter and stones, algae, drying, among others) at the outlet of the reactor. The gas produced by digestion is known as biogas. It has a composition of approximately 40–70% methane gas ($CH_4$) and 29–59% carbon dioxide ($CO_2$) with negligible traces of oxygen and hydrogen sulfide ($H_2S$), which gives it a very distinctive smell, which is beneficial in identifying leaks in the system.

$$\frac{dS}{dt} = -\left(\frac{\mu_{max}(X_T, S)}{Y_{sx}}\right) + D(S_0 - S) \tag{1}$$

$$\frac{dX_T}{dt} = \left(\frac{\mu_{max}(X_T, S)}{Y_{xs}}\right) - F_d X_T - D X_T \tag{2}$$

$$\frac{dCO_2}{dt} = \mu_{max}(X_T, S) Y^\tau Y_{xco_2} CO_2^\alpha - DCO_2 \tag{3}$$

$$\frac{dCH_4}{dt} = \mu_{max}(X_T, S) Y^\omega Y_{xch_4} CH_4^\beta - DCH_4 \tag{4}$$

where:

$$\begin{aligned}
\mu_{\max}(X_T, S) &= \ Growth\ rate \\
Y_{sx} &= Substrate - Biomass\ yield \\
Y_{xs} &= Biomass - Substrate\ yield \\
CO_2 &= Carbon\ dioxide \\
D &= Dilution\ rate \\
F_d &= Death\ cell\ dynamics\ rate \\
\beta &= Order\,parameter \\
\alpha &= Order\,parameter \\
Y &= conversion\,factor
\end{aligned} \tag{5}$$

Based on the matter balance, for a continuous rector to simulate the behaviour of the substrate, biomass and gases, in addition, it uses the chemical reactions that occur within the biodigester as shown in the equation:

$$DQO + H_2O + X_T \longrightarrow H_2O + CH_4 + CO_2 + H_2S + X_T \tag{6}$$

Considerations for Equations (1)–(4):

(i)     The Anaerobic Digestion Model No.1 (ADM1) is a generalized mathematical model for the AD process that describes the biochemical and physicochemical processes that occur in a biodigester. To carry out the mathematical modeling, it is necessary to start from the basis that the AD process is composed of sub-processes: hydrolysis, acidogenesis, acetogenesis, and methanogenesis. In this work, we propose the use of a simplified mathematical model of ADM1 considering the following biochemical reactions for engineering, optimization and process control purposes.

(ii)

$$Complex - process \longrightarrow simplified - reaction - scheme \tag{7}$$

(iii)　Acidogenesis process

$$S_1 + H_2O \longrightarrow H_2O + CO_2 + X_1 \tag{8}$$

(iv)　Methanisation process

$$S_2 \longrightarrow CH_4 + CO_2 + X_2 + H_2S \tag{9}$$

hence

- Organic matter: $S_1$
- Acidogenic bacteria: $X_1$
- Fatty volatile acids: $S_2$
- Methanogenic bacteria: $X_2$
- $X_T : f(X_1 X_2)$
- $S : f(S_1 S_2)$

Based on publications such as [5], we offer a mathematical model in which the biomass variable represents the total of the whole consortium of acidogenic and methanogenic bacteria. For the construction of the reaction kinetics, modified Haldane structure was taken as a basis [28], which describes a state variable's change within the biodigester. Therefore, $\mu_{max}(S, X)$ is defined above and the general model is described below:

$$\mu_{max}(S, X_T) = \left[ K_{max} \frac{S}{K_s + S + K_i} X_T \right]^{\phi} \tag{10}$$

The Haldane equation is a mathematical model of inhibition widely used in biotechnology and describes microorganisms growth in an aqueous environment, which is limited by the substrate and some other metabolite [29,30]. Where:

$K_{max}$ = Substrate constant degradation
$K_s$ = Mass coefficient in the substrate aqueous medium
$K_i$ = Inhibition factor

The proposed mathematical model proposed in this work maintains far fewer parameters and various other parts of it, hence it is easier to implement the state estimators for complex variables such as the concentration of methane, biomass, or substrate (see Figures 1 and 2).

AD processes can be referred to as wet and dry digestion, depending on the feeding substrate's total solids concentration. AD is defined as a wet process if the substrate's total solids concentration is less than 15%, and as a dry process if the concentration lies within the 20–40% range. In wet processes, the solid waste must be conditioned to the appropriate solids concentration by adding process water either by re-circulation of the liquid effluent fraction or by digestion with more liquid waste. The latter is a desirable method for combining various waste streams such as sewage sludge or manure and the organic fraction of municipal solid waste [31]. The reactors used in wet AD processes are generally called continuous stirred tank reactors (CSTR), applying mechanical mixers or a combination of automatic mixing and biogas injection [32]. The application of a wet AD process offers several advantages, such as diluting substances using the process water and the need for less sophisticated mechanical equipment. However, disadvantages such as complicated pretreatment, high-water and -energy consumption for heating, and reducing the working volume due to sedimentation of inert materials must be considered [33].

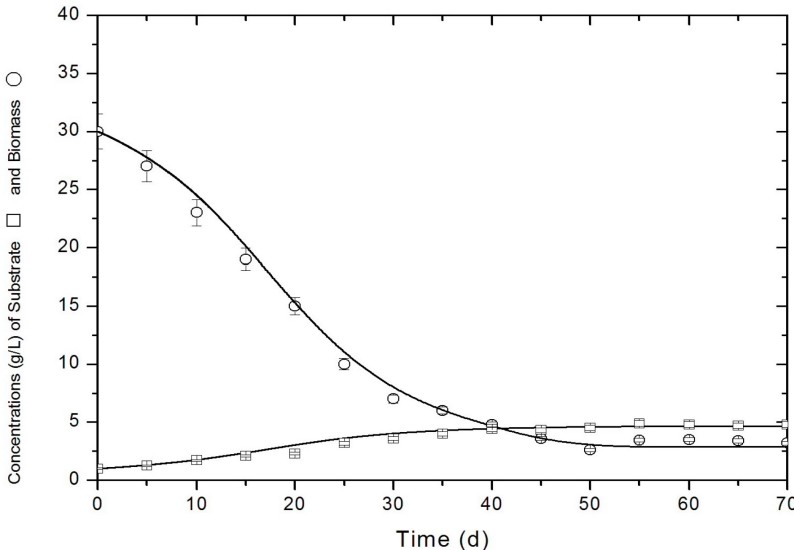

**Figure 1.** Real data (symbol) and predicted dynamics (continuous line) of substrate-biomass.

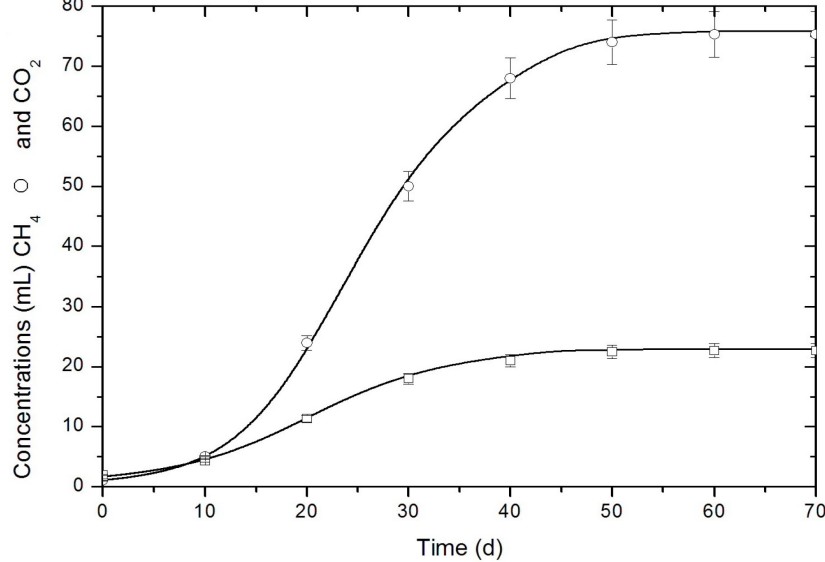

**Figure 2.** Real data (symbol) and predicted dynamics (continuous line) of $CH_4$-$CO_2$.

### 3. Observer Design and Observability Analysis

The state observer provides indirect measurements (which, by themselves, are not as easy to measure online physically) under real conditions by combining data detected from a group of heterogeneous physical sensors (inline). Signals from the individual sensors can be used in calculations within a single-variable estimator state of reconstruction, in addition to deviations between the trajectories of a real system and the predictions of a mathematical model. A virtual sensor is constructed from a state observer comprising a simulation or measurement of an online sensor (real-time) of the system or plant coupled to the plant model, driven by a correction term derived from the difference between the system's actual output and the expected output derived from the observer. Therefore, it introduces an alternative state observer, who can converge to the states simulated with satisfactory performance. To this end, the following structure represents the proposed observers and their corresponding convergence analysis. The condensing of (1–4) as single-input and single-output (SISO) system is proposed as follows for $u = D$ and $\mathbb{R}^n$:

$$\dot{x} \quad = \quad f(x,u) + \underbrace{\delta(u,x)}_{\delta(\cdot)} \tag{11}$$

$$
\begin{aligned}
x(0) &= x_0 \\
y &= h(x) = Cx \\
C &= [1\,0\,0\ldots0] \tag{12}
\end{aligned}
$$

where $x^T = [x_1, x_2, \ldots x_n]^T \in \mathbb{R}^n$, $y \in \mathbb{R}$, $C \in \mathbb{R}^{1xn}$, $u(t) \in \mathbb{R}$. The disturbance $\delta(\cdot)$ is the unknown disturbance function. On the stated dynamics $\delta(\cdot)$, parameter changes and their consequences on system dynamics may be represented and found.

**Assumtions 1.** *It is proposed that there is a vector called estimated $\hat{x}$, such that there are arithmetical differences such that $\epsilon = \hat{x} - x$*

*Therefore, the functions $f(x,u)$ and $f(\hat{x},u)$ are a Lipschitz function such that:*

$$\left\| \underbrace{f(x,u) - f(\hat{x},u)}_{\phi(\cdot)} \right\| \leq l_1 \|\epsilon\|$$

*where $\epsilon \in \mathbb{R}^n = x - \hat{x}$ is the estimating error vector, while $C\epsilon = e_0 \in \mathbb{R}$ with $e_0$ as Output Signal Error (OSE).*

**Assumtions 2.** *There is a constant $k_2 > 0$ and OSE such that the system following condition is fulfilled for (11):*

$$\left| \frac{k_2 e_0}{1 + e_0} \right| \leq 1$$

*with the above, the following proposal proposes a non-linear logistics estimator.*

**Assumtions 3.** *The nonlinear unknown function $\delta(\cdot)$ is a bounded function such that:*

$$
\begin{aligned}
\|\delta(\cdot)\| &< k \tag{13} \\
\forall k &> 0
\end{aligned}
$$

**Note 1** It is highly typical in biological systems for this type of system to contain uncertainties, especially concerning possible changes in the control input. This is because in this type of system, the $u(t)$ modifies the operating parameters in the stationary phase. As a result, the delta perturbation might be asked to be minimum constrained. A preferable case would be for this distortion to be Lipschitz to the OSE, which would result in:

$$
\begin{aligned}
\|\delta(\cdot)\| &\leq l_2 e_0 \\
\forall l_2 &> 0
\end{aligned}
$$

The above would be an ideal case since it gives us more information about the perturbation $\delta(\cdot)$. However, it is often difficult to carry this out, even the calculation of the $l_2$ is not trivial. Therefore, in this work we propose to work with the only known fact that the perturbation is bounded as in Equation (13).

**Theorem 1.** *Let system (11) be an SISO system that complies with the assumptions. Then, the following NLLO observer is proposed:*

$$\dot{\hat{x}} = f(\hat{x}, u) + K_1 e_0{}^p \left( \frac{k_2 e_0}{1 + e_0} \right) + \delta(\cdot) \tag{14}$$

*where $k_1 > 0$ is a vector gain. Therefore, the dynamics error $\epsilon = x - \hat{x}$ is said to be stable and to converge to a final level. For the vector gains $K_1 > 0$ and the scalar $k_2 >> p + 1$ and $p >> 2$*

**Proof.** Let us consider the dynamics of the estimation error.

$$\dot{\epsilon} = \dot{x} - \dot{\hat{x}}$$

$$\dot{\epsilon} = f(x, u)$$

$$- \left[ f(\hat{x}, u) + K_1(Cx - Cx)^P \left( \frac{k_2(Cx - Cx)}{1 + |Cx - Cx|} \right) \right] + \delta(\cdot)$$

$$\dot{\epsilon} = \underbrace{f(x, u) - f(\hat{x}, u)}_{\phi(\cdot)} - K_1 e_0^p \left( \frac{k_2 e_0}{1 + e_0} \right) + \delta(\cdot)$$

$$\dot{\epsilon} = \phi(\cdot) - K_1 e_0^p \left( \frac{k_2 e_0}{1 + e_0} \right) + \delta(\cdot)$$

It is proposed that a Lyapunov function and its derivate

$$V = \frac{1}{2} \epsilon^T \Gamma \epsilon \tag{15}$$

$$\dot{V} = \dot{\epsilon}^T \Gamma \epsilon + \epsilon^T \Gamma \dot{\epsilon}$$

$$\dot{V} = \left( \phi^T(\cdot) - K_1^T e_0^p \left( \frac{k_2 e_0}{1 + e_0} \right) + \delta(\cdot) \right)^T \Gamma \epsilon$$

$$+ \epsilon^T \Gamma \left( \phi(\cdot) - K_1 e_0^p \left( \frac{k_2 e_0}{1 + e_0} \right) + \delta(\cdot) \right)$$

where $\Gamma > 0$ symmetric matrix. Therefore, it is majored:

$$\dot{V} \leq 2\|\phi(\cdot)\|^T \|\Gamma\| \|\epsilon\| - 2K_1^T e_0^p \left| \frac{k_2 e_0}{1 + e_0} \right| \|\Gamma\| \|\epsilon\|$$

$$+ 2\|\delta\|^T(\cdot)\|\Gamma\| \|\epsilon\| \tag{16}$$

$$\text{where } \|\Gamma\| = \lambda_{\max}\{\Gamma\} = a_1$$

$$\|\phi(\cdot)\|^T = \|\phi(\cdot)\| = l_1 \|\epsilon\|$$

$$\|\delta\|^T = \|\delta\| = k \tag{17}$$

$$\|K\|_1^T k_2 = a_2 \tag{18}$$

Therefore:

$$\dot{V} \leq 2a_1 \|\epsilon\|^2 - a_1 a_2 \underbrace{e_0^p \left| \frac{k_2 e_0}{1 + e_0} \right|}_{\leq e_0^p} \|\epsilon\| + 2a_1 k \|\epsilon\|$$

$$\dot{V} \leq 2a_1 \|\epsilon\|^2 - a_1 a_2 \left\| \underbrace{C\epsilon}_{e_o} \right\|^p + 2a_1 k \|\epsilon\|$$

Via Cauchy–Schwartz inequality, the above term can be maximised in the following way:

$$\dot{V} \leq 2a_1 \|\epsilon\|^2 - 2a_1 a_2 \|C\|^p \|\epsilon\|^p + 2a_1 k \|\epsilon\|$$

$$\dot{V} \leq 2a_1 \|\epsilon\|^2 - 2a_1 a_2 \underbrace{\|C\|^p}_{\leq 1} \|\epsilon\|^p + 2a_1 k \|\epsilon\|$$

$$\frac{\dot{V}}{2a_1} \leq \|\epsilon\| (\|\epsilon\| + k - a_2 \|\epsilon\|^{p-1})$$

It is easy to observe that for $a_2 > k$ and $p >> 2$, it has $a_2 \|\epsilon\|^{p-1} > \|\epsilon\| + k$. Therefore:

$$V \leq -2a_1 \beta(\epsilon, p, a_2, k)$$

where

$$(2a_1)^{-1}\beta(\epsilon, p, a_2, k) =$$
$$3^{-1}\|\epsilon\|^3 + 0.5k\|\epsilon\|^2 - a_2(p+1)^{-1}\|\epsilon\|^{p+1}$$

Hence, since the $\beta(\epsilon, p, a_2, k)$ is an unknown function but increases in error infinitely, and is also a smooth polynomial function that depends on the degree $p$, value of $k$ and $a_2$, the system is hence stable. □

**Note 2** The influence of the parametric change owing to the system's parameters, especially biological conditions, is within the convergence of the observer's system, as in cases where the unknown value k of Equation (13) would change, denoting a change in the conditions of the function $\beta(\epsilon, p, a_2, k)$ as previously defined. Given the robustness qualities, it may be concluded that the observer will be resilient in the presence of parametric change.

## 4. Biodigester Numerical Results and Observability Analysis

Asymptotic observer and observability analysis, as mentioned above, cannot be built without first evaluating the mathematical model using experimental data from [34,35]. Biomass production, biogas release, and substrate use were all confirmed to occur at the projected rates revealing the estimated timeframe within which the estimator can function. To analyze which sensors will be most effective during the state-estimation phase, we will use the results of the simulated experiments to select a stationary data set and then compute the linear matrices A, B, and C required to determine the local observability at this equilibrium point and verify the observability concerning each of the four state variables (see Figures 3–6) (Table 1).

**Table 1.** Simulation parameters obtained from the literature [34,35].

| Parameters | Value | Range |
|:---:|:---:|:---:|
| $K_s$ | 150 g/L $\pm$ 15 g/L | 50–246 g/L |
| $K_i$ | 50 g/L $\pm$ 15 g/L | 192 |
| $F_d$ | 0.009 d$^{-1}$ $\pm$ 0.0019 d$^{-1}$ | 0.0005–0.0124 d$^{-1}$ |
| $S_0$ | 30 g/L | 5.057 g/L |
| $Y_{sx}$ | 0.426 $\pm$ 0.21 | 0.2–0.80 |
| $Y_{xs}$ | 0.333 $\pm$ 0.11 | 0.3–0.90 |
| $D$ | 0.001 $d^{-1}$ | 0.015 $d^{-1}$ |
| $\alpha$ | 0.29 $\pm$ 0.18 | 0.1–1.9 |
| $\beta$ | 0.20 $\pm$ 0.09 | 0.1–1 |
| $Y_{xco_2}$ | 0.67 | 0.1–1 |
| $Y_{xch_4}$ | 0.78 | 0.1–1 |
| $\mu_{max}$ | 0.23 g/Ld | 0.1–1.5 d$^{-1}$ |
| $Y$ | 1.2 mL*(L/g) | 0.1–1.5 |
| $\phi$ | 1.9 mL*(L/g) | 0.1–2.5 |

Experimental values from [34,35] were used to verify the results of the numerical simulations in order to obtain a rough idea of the biomass, biogas production, and substrate consumption that can be expected in the real world and thus what sort of range the estimator can be expected to work in. Due to the lengthy nature of AD operations, the simulation ran over a 70-day period. When biogas output is constant, the system stabilizes after around 60 days. The linearization process described below will begin with these steady-state values. Similar comparisons were performed between the substrate and biomass variables depicted in the figure. A variety of Grammy matrices with local observability can be analyzed via equilibrium points using the linearization analysis of Jacobian matrices. Initial conditions for a time-dependent analysis: substrate 30 g/L, 1 g/L Biomass, 1.1 g/L Carbon Dioxide, and 1.7 g/L Methane, to $D = 0.001d^{-1}$.

$$\dot{x}_{lin} = Ax_{lin} + Bu_{lin}$$
$$y = C_{lin}x_{lin}$$

Therefore, the rank > n is as follows:

$$O_{lin} = \begin{bmatrix} C_{lin} \\ C_{lin}A_{lin} \\ \vdots \\ C_{lin}A_{lin}^{n-1} \end{bmatrix}$$

The system and this steady state are observable for different outputs for $C = C_{lin}$, such that it shows:

- $C = \begin{bmatrix} 1 & 0 & 0 & 0 \end{bmatrix}$
  If the substrate is measured (S)
- $C = \begin{bmatrix} 0 & 1 & 0 & 0 \end{bmatrix}$
  If the biomass is measured ($X_T$)
- $C = \begin{bmatrix} 0 & 0 & 1 & 0 \end{bmatrix}$
  If the carbon dioxide is measured ($CO_2$)
- $C = \begin{bmatrix} 0 & 0 & 0 & 1 \end{bmatrix}$
  If the methane is measured ($CH_4$)
- $C = \begin{bmatrix} 0 & 0 & 1 & 1 \end{bmatrix}$
  If the biogas is measured ($CH_4$ y $CO_2$)

Table 2 demonstrates how the observability of a system can be indicated by the impact of employing a particular sensor. Biogas sensors ($CO_2$ and $CH_4$) make it straightforward to estimate factors such as biomass and substrate that are notoriously difficult to measure with analog or digital sensors. As a result, this observer-based sensor software (with $CO_2$ sensors) offers a novel and practical solution to the issue of actual biodigester monitoring. This is based on the linearized model's local observability data close to the dilution cup's equilibrium point. The design of a state estimation robust to changes in the dilution rate is crucial for achieving a strong adequate state estimator, as a change in the dilution rate affects all equilibrium and estimation conditions. The authors propose measuring $CO_2$ as a sensor that already exists on the market and using it to estimate local variables such as biomass, substrates (nutrients), and methane. Estimated adversaries are $\hat{x} = [43\ 4.1\ 2.1\ 5]^T$, and the initial conditions are $x_0 = [30\ 1\ 1.7\ .1]^T$. The performance of the proposed observer (refobserver) was compared to that of a Luenberger observer and a sliding mode (See, Figures 3–6).

**Table 2.** Comparative analysis of estimable variables [36].

| | | | Online Estimations | | |
|---|---|---|---|---|---|
| Case | Variable | $S$ | $X_T$ | $CO_2$ | $CH_4$ |
| [1 0 0 0] | $S$ | ● | ○ | ● | ○ |
| [0 1 0 0] | $X_T$ | ○ | ● | ● | ● |
| [0 0 1 0] | $CO_2$ | ● | ● | ● | ● |
| [0 0 0 1] | $CH_4$ | ● | ● | ● | ● |
| [0 0 1 1] | $CO_2$, $CH_4$ | ● | ● | ● | ● |

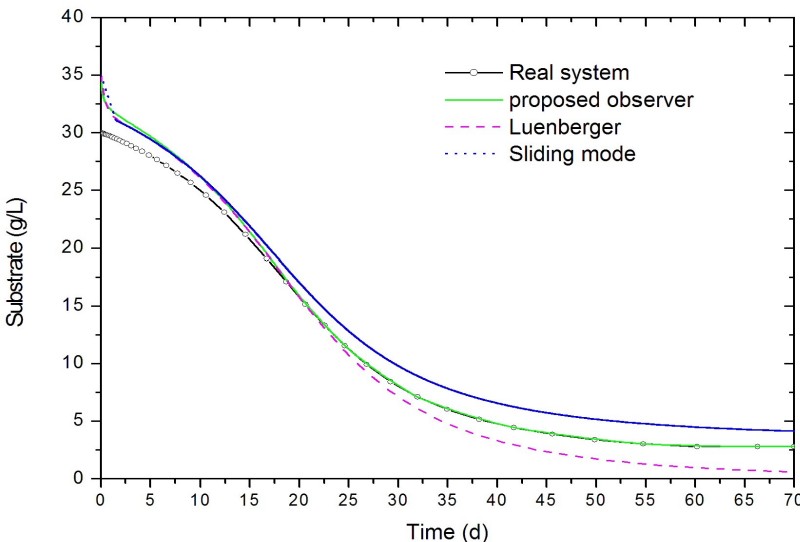

**Figure 3.** Simulation and comparison of the continuous-time observers, proposed observer vs. real system: Dynamics of the substrate.

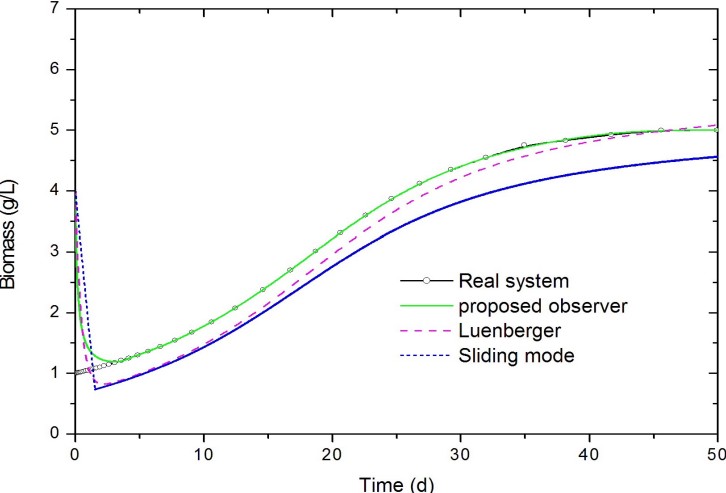

**Figure 4.** Simulation and comparison of the continuous-time observers, proposed observer vs. real system: Dynamics of the biomass.

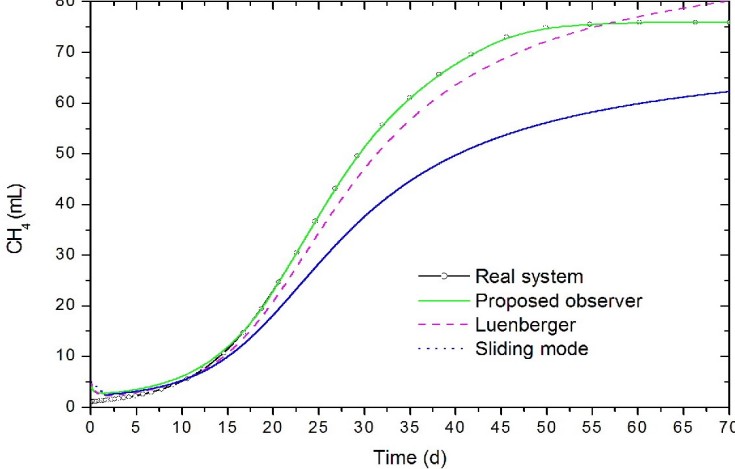

**Figure 5.** Simulation and comparison of the continuous-time observers, proposed observer vs. real system: Dynamics of the $CH_4$.

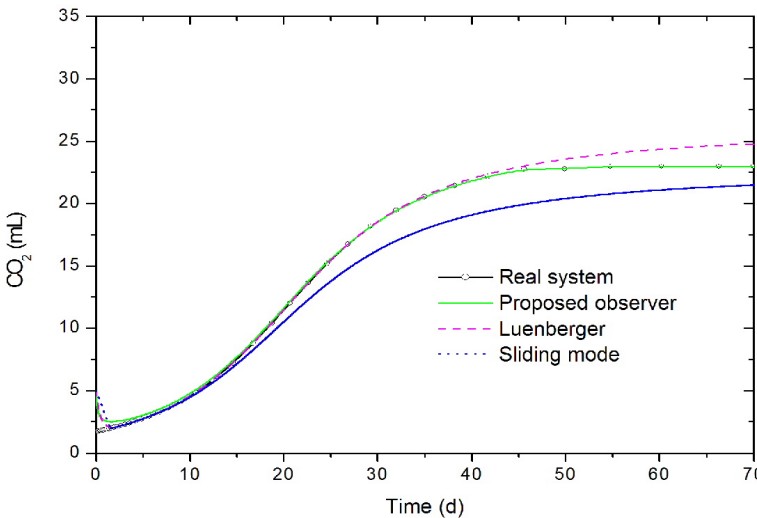

**Figure 6.** Simulation and comparison of the continuous-time observers, proposed observer vs. real system: Dynamics of the $CO_2$.

### 4.1. Analysis of Results

To begin, the suggested estimator outperforms the sliding mode estimator, which has an Integral Absolute Error (IAE) of 695, and the classical Luenberger observer, which has an IAE of 300, when measured with IAE index. ITAE (Time-weighted Absolute Error) is displayed after the comparison is made. Similarly, the proposed estimator outperforms the rest, and is therefore chosen as the ideal solution for this case study, which also shows that the more time that passes, the better the solution is.The proposed estimator outperforms the alternatives, and is hence chosen as the optimal solution. There is a correlation between performance level and time spent on stage. The sliding mode observer (ITAE = 550), the Luenberger observer (ITAE = 35), and the best-recommended observer (ITAE = 5) all have significantly larger ITAE calculations than the others. Accordingly, the proposed observer will be used in the experiment's follow-up phase, where it will be emulated in advance of changes to the operating parameters by proposing dilution rate perturbations.

### 4.2. Robustness Disturbance Parameters Input Signals

The input paramters can be measured and quantified; however, it is impossible to empirically establish whether the internal metabolic parameters have changed. A simulation study is provided on the basis of the identical analytical circumstances as in the preceding section, except for suggesting a dynamic disturbance in the dilution rate signal to illustrate the system resistance to these types of input parameter changes, which are typical in biological systems. In order to show the performance of the observer under output disturbances, it was assumed that the reactor was operated in semi-batch mode, with the dilution rates varying within a range D = (0.01 to 0.05) $d^{-1}$, as shown in Figure 7. The proposed observer provides a good estimation of unknown states (Figures 8 and 9). The following diagrams show the perturbation dynamics. For application purposes, the $CO_2$ concentration was considered as the measured output of the observers; this is justified because this concentration is one of the easier to measure of the biodigester variables via a low-cost senspr, and also because of the observability analysis results (See, Table 3).

**Table 3.** Comparison of estimation methods applied in AD processes [37–44].

| State Estimator | Available Variables (Online) | Reconstructed Variables | Advantages | Disadvantages |
|---|---|---|---|---|
| Sliding- mode estimators | $CH_4$ | Volatile fatty acid (VFA) concentration | Rrobustness to disturbances and unmodeled dynamics | Steady-state error |
| Neural networks | Biomass | Substrate | You need no prior knowledge about kinetic growth rate | You need experimental data to properly train the neural network |
| Kalman filtering algorithm | Biomass | Substrate | It strongly depends on the precision of the model, numerical problems and difficulty of convergence | High computational cost |
| Adaptive observer | Specific growth rate and cell concentration | Substrate | Does not require any type of analytical description of the specific growth rate | You need a proper theoretical analysis of the properties of the model |
| Proposed estimator | $CH_4$ or $CO_2$ | Biomass, and Substrate | The observer input is an in-line sensor and simple to implement | Depends on the model It is not robust against modeling errors |

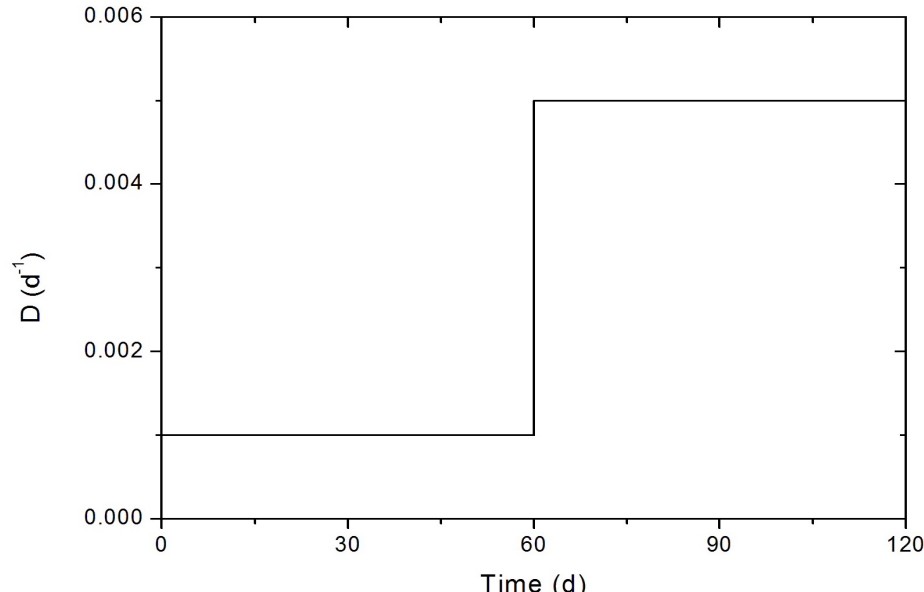

**Figure 7.** Parametric change on the proposed parameter D.

To study the performance of an observer and its robustness, it is necessary to put it in the presence of abrupt perturbations as shown in Figure 8.

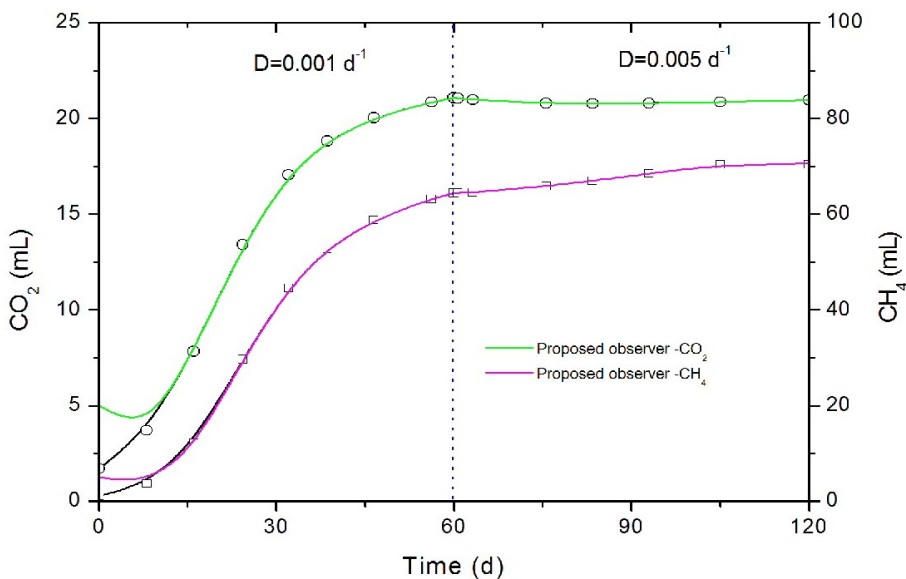

**Figure 8.** The $CO_2$ and methane dynamics in the presence of abrupt disturbance to 60 days on process.

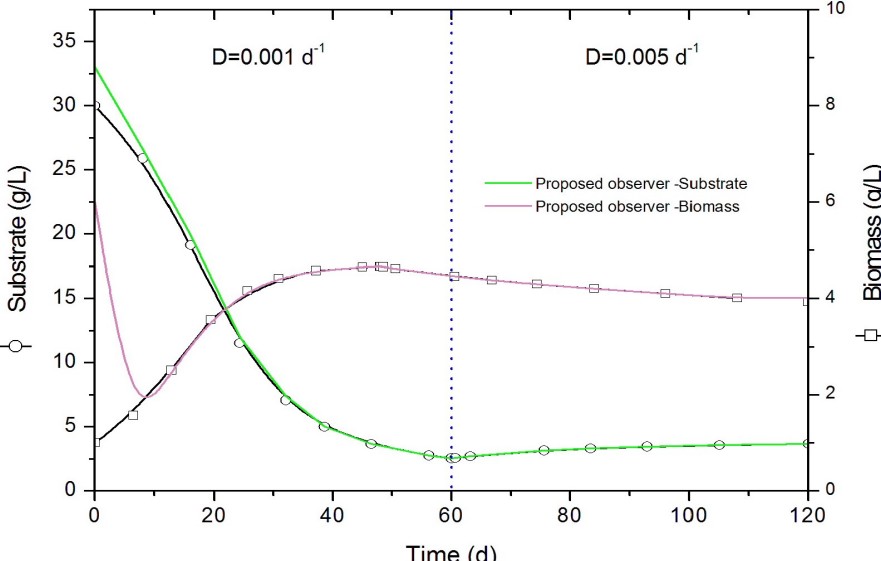

**Figure 9.** Dynamics of the biomass and nutrients in presence of abrupt disturbance to 60 days on process.

Finally, it is important to take into account that the AD system may have external disturbances, since these can occur naturally in this type of application, the above may be due to a damaged sensor, poor connections or interference in sampling, for which the simulation was carried out for the case in which the disturbance occurs in the $CO_2$ sensor, as can be seen in Figure 10 and described by Equation (19):

$$\frac{dCO_2}{dt} = \mu_{max}(X_T, S)Y^\tau Y_{xco_2}CO_2^\alpha$$
$$+CO_2 sin(0.20\pi) + 20cos(10.9\pi) - DCO_2 \tag{19}$$

Given the rather sluggish dynamics after 60 days, we suggest a modification illustrated in Figure 10 that must be corrected by the estimator in order to get a fine and robust estimation of the states recorded with the soft sensor proposed by the NLLO observer.

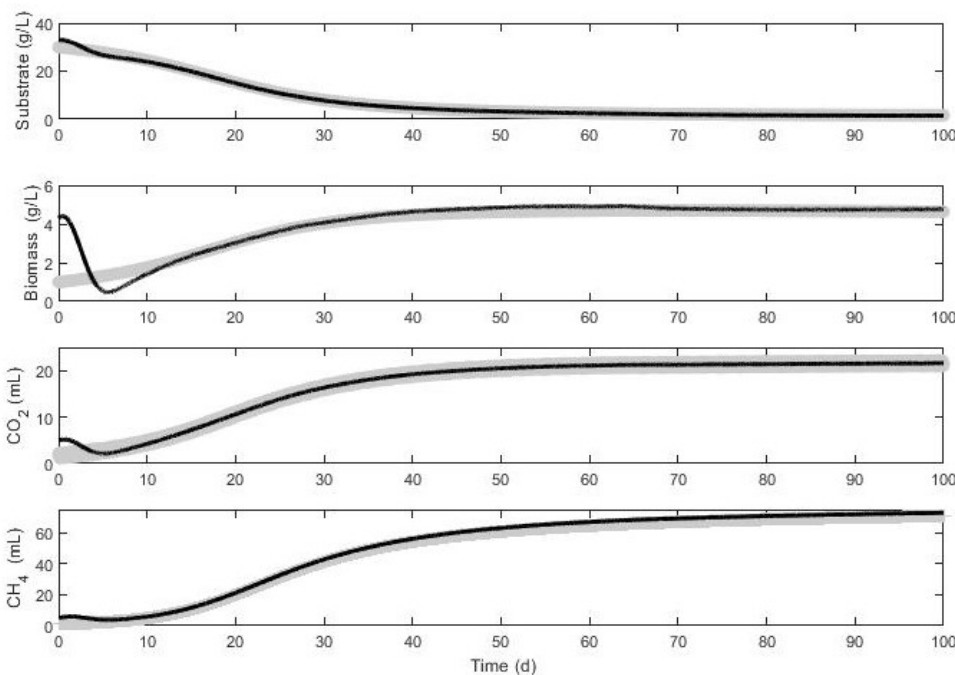

**Figure 10.** Simulation of disturbances considering noise in the $CO_2$ sensor.

## 5. Conclusions

In this paper, a kinetic model was created to offer an experimental prediction for ranges in biogas production reported in the literature in continuous operations. Additionally, a local observability analysis was conducted, in which multiple combinations of output measures were used to indicate the dimensions of the associated observable subspaces using the observability matrix criteria. The state estimator is built on the understanding of a single measurable variable in the line of sight. The state estimator is based on the knowledge of a single measurable quantity, online, $CO_2$ concentration, which can be used to estimate substrate, biomass, and methane concentrations. A combination of substrate, biomass, and methane numerical simulation is used to verify the stability and asymptotic convergence. Numerical simulation using a performance index verifies the stability and asymptotic convergence. Finally, it is clear that this method allows for rapid calculation of a mathematical description of continuous fermentation processes that can be utilized for optimization and control. This work may be expanded to situations with mismatched parameters and dispersed delays, and advanced controllers can be built, such as Finite-Time Control of Dual-Switching Poisson, as in [45,46], offering a unique application for chemical–biological systems; however, these avenues remain yet unexplored.

**Author Contributions:** Conceptualization, C.A.L.-C., E.G.-V. and P.A.L.-P.; methodology, P.A.L.-P.; software, E.G.-V.; validation, C.A.L.-C., E.G.-V. and P.A.L.-P.; formal analysis, A.E.R.-M. and P.A.L.-P.; investigation, E.G.-V. and A.E.R.-M.; resources, J.A.M.-H.; writing—original draft preparation, C.A.L.-C., A.E.R.-M., P.A.L.-P. and R.B.-A.; writing—review and editing, A.E.R.-M., P.A.L.-P., J.A.M.-H. and R.B.-A.; supervision, P.A.L.-P.; project administration, P.A.L.-P. All authors have read and agreed to the published version of the manuscript.

**Funding:** Convocatoria 2023: Proyectos de Investigación Científica, Desarrollo Tecnológico e Innovación of Tecnólogico Nacional de Mexico.

**Institutional Review Board Statement:** Not applicable.

**Informed Consent Statement:** Not applicable.

**Data Availability Statement:** Not applicable.

**Acknowledgments:** CONACYT for the Master's scholarship awarded to E.A.G.-V. with registration number 902332.

**Conflicts of Interest:** The authors declare no conflict of interest.

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
