# Peer review of "State Estimation in a Biodigester via Nonlinear Logistic Observer: Theoretical and Simulation Approach"

_processes, doi:10.3390/pr11041234_

Round 1

Reviewer 1 Report

Remarks:

1 1 The mathematical model used to design a process state observer is totally unacceptable. It involves only one microbial population and one substrate, which could be applied to the simplest biotechnological process, but not to the Anaerobic Digestion (AD). The models known for this process include at least 2 populations (acidogens and methanogens) and 2 substrates. It is mandatory to include hydrolysis, which is a key stage of AD. Therefore, further work becomes meaningless.

2.     2.  In the literature used, 71% of the references are older than 5 years.

3.    3.  Ignorance of the process has also led to ignorance of the specific terminology - mixing “digester” and “digestion”, “sludge” and “digestate”, “central bacterial population” (page 2, line 34), – non existing term etc.

4.     4.  Bad style and frivolous layout of the article ("0. Introduccion", lack of "References", etc.)

5.    5.   Ignorance of the basic literary sources in the field of AD also led to incorrect citation of the references used.

Author Response

In the PDF I will send you all the clarifications and corrections on your comments, I send you my best regards.

Reviewer 2 Report

The paper proposes an NLLO fos state reconstruction in an anaerobic digestor nonlinear model. The proposed observer intends to achieve performances that outrange the exponential and the asymptotical designs by taking the best features of these two approaches.

The paper is well written, and the results illustrate the proposed observer's benefits in the anaerobic digestor process context.

However, some minor issues must be solved to improve readability.

1. On the writing, consider the following:

(i) Some phrases/statements seem incomplete or lack sense. As instances:

"- Hybrid observers: The basic idea behind hybrid observers is to combine the advantages of exponential and asymptotic observers. And the mixed observer is to combine the benefits of exponential and asymptotic observers [20].", is there some difference between hybrid and mixed observers?

"Hence, the system is stable, β(ϵ, p, a2, k) is an unknown function, but it is infinitely increasing in error and is also a smooth polynomial function which depends on the degree p, value of k and a2, hence the system is stable.", at the beginning of this statement, something as 'Since that β(ϵ, p, a2, k) is an unknown...' may sound better?

(ii) In the stability/convergence proof of the proposed observer, several hats, as in f, x, are absent (see below).

(iii) Please check some captions for tables. Some of them are in the Spanish language.

2. Regarding the proposed observer, why not opt for a discrete-time one? Please, explain it in the revision notes or the text.

Author Response

(The authors gave the same response as above.)

Reviewer 3 Report

This paper proposed an analytical design of a non-linear logistic observer and a performance comparison with notable state estimators to predict the state variables in a biodigester. The subject discussed in the paper seems to be interesting. However, there are many important issues that need to be discussed further, so I think the present version should be revised carefully. Some of the revision suggestions are listed as follows:

1. Compared with the existing related papers, what are the essential contributions of this work? What is the restriction of this paper? The authors can explain it clearly;

2. The authors require to study the effect of parameter variation for the mathematical model on the behavior of the proposed novel NLLO observerï¼›

3. What is the practical significance of studying the novel NLLO observer in this paper? The authors can combine the actual application examples for specific analysis in the biodigester numerical results and observability analysis.

4. According to the topic of the paper, the authors may propose some interesting problems as future work in Conclusion. The following references can help enrich future research. Dynamic self-triggered impulsive synchronization of complex networks with mismatched parameters and distributed delay; Positiveness and finite-time control of dual-switching Poisson jump networked control systems with time-varying delays and packet drops.

Author Response

(The authors gave the same response as above.)

Round 2

Reviewer 3 Report

The authors gave a better revision. Now, the revised work may be published in this journal.